# Positron Emission Tomography in Coronary Heart Disease

José de Almeida [1,2,*] ![ID], Sofia Martinho [1] ![ID], Lino Gonçalves [1,2] and Maria Ferreira [1,2,3,*] ![ID]

1 Department of Cardiology, Coimbra Hospital and University Centre (CHUC), 3004-561 Coimbra, Portugal; sofia14martinho@gmail.com (S.M.); lgoncalv@ci.uc.pt (L.G.)
2 Faculty of Medicine, University of Coimbra (FMUC), 3004-561 Coimbra, Portugal
3 Institute for Nuclear Sciences Applied to Health (ICNAS), 3004-561 Coimbra, Portugal
* Correspondence: josepauloalmeida92@gmail.com (J.d.A.); mjvidigal19@gmail.com (M.F.)

**Abstract:** With advances in scanner technology, postprocessing techniques, and the development of novel positron emission tomography (PET) tracers, the applications of PET for the study of coronary heart disease have been gaining momentum in the last few years. Depending on the tracer and acquisition protocol, cardiac PET can be used to evaluate the atherosclerotic lesion (plaque imaging) and to assess its potential consequences—ischemic versus nonischemic (perfusion imaging) and viable versus scarred (viability imaging) myocardium. The scope of this review is to summarize the role of PET in coronary heart disease.

**Keywords:** positron emission tomography; coronary heart disease; plaque imaging; perfusion imaging; viability imaging; 18F-sodium fluoride; 18F-fluorodeoxyglucose; Rubidium-82; 13N-ammonia; 15O-water

## 1. Introduction

PET has been increasingly used in the medical field since its discovery, in the sixth decade of the last century, due to both technological advances and the expansion of indications. Fifty years after its development, the number of detectors increased from 64 to 19,000 and the spatial resolution of the exam from 14 mm to 4 mm [1]. These detectors measure the total energy deposited by the two annihilation photons moving in opposite directions produced after positron emission from a radionuclide-tagged tracer molecule [2]. The tracer molecule is part of the biochemical process that we are interested in quantifying and as such, the radionuclide should ideally only interact with that molecule [3]. The raw data acquired is then corrected for attenuation and finally reconstructed to provide an estimate of the in vivo tracer distribution [2].

The evolution of PET has had a huge impact in the study of cardiovascular disease, including coronary heart disease, heart failure, cardio-inflammatory disease, valvular heart disease, and assessment of cardiac devices and cardiac tumors.

Coronary heart disease is defined as an inadequate blood flow to an area of myocardial tissue due to blockage of the blood vessels that supply it, most often by an atherosclerotic plaque [4].

PET can be used to evaluate the atherosclerotic lesion (plaque imaging) and to assess its potential consequences—ischemic versus nonischemic (perfusion imaging) and viable versus scarred (viability imaging) myocardium [5].

## 2. Plaque Imaging

Although coronary atheromatous plaques are often present in older patients, in most cases their development is silent and will not result in a cardiovascular event. In contrast, some atherosclerotic plaques will rupture and cause myocardial infarction. Identifying these "vulnerable" plaques before they rupture and differentiating them from their "stable" counterparts is therefore a key objective for a cardiac imaging technique [6].

PET scanners have the capacity to assess any biochemical process occurring in the atherosclerotic plaque, if the appropriate surrogate tracer molecule is used.

## 2.1. Imaging Inflammation with 18F-Fluorodeoxyglucose (FDG)

FDG is overall the most used radioligand in PET imaging. It is a glucose analog that enters cells via facilitated glucose transporter member (GLUT) 1 and 3 and undergoes phosphorylation to become (18)F-FDG-6-phosphate, which cannot exit the cell before radioactive decay. Therefore, signal intensity correlates with cellular glucose uptake and phosphorylation, which indicates high metabolic activity. In the case of the atherosclerotic plaque, metabolic activity appears to be related to the concentration of proinflammatory macrophages, as demonstrated in animal models [7]. In humans, serum levels of myeloperoxidase were associated with carotid plaque FDG uptake [8].

Aortic FDG uptake was independently associated with cardiovascular risk factors such as increased low-density lipoprotein and total cholesterol, and with the presence of metabolic syndrome [9–11]. On average, patients with myocardial infarction (MI) were shown to have higher aortic FDG uptake compared with stable angina patients [12]. Visceral adipose tissue uptake of FDG was associated with systemic inflammatory status, and with the presence of metabolic syndrome components [13]. Additionally, it was independently associated with the severity of CAD and with the occurrence of AMI [14].

Regarding the assessment of pharmacological treatment, statins were shown to reduce the arterial FDG signal in a dose-dependent manner and pioglitazone was shown to attenuate vascular FDG uptake [15–17]. In contrast, the novel antidyslipidemia drugs Dalcetrapib (cholesteryl ester transfer protein inhibitor) and Rilapladib (lipoprotein-associated phospholipase A2 inhibitor) did not have any effect on vascular FDG activity [18,19].

There is, however, no current evidence to support the implementation of FDG-PET for atherosclerotic risk stratification in clinical practice.

## 2.2. Imaging Microcalcification with 18F-Sodium Fluoride (NaF)

NaF has been used to detect bone metastases and it is known to replace the hydroxyl group of hydroxylapatite in areas of calcification [20,21]. Regarding vascular calcification, two processes occur in a continuum in the atherosclerotic plaque as a healing response to the necrotic core inflammatory mediators: microcalcification, followed by macrocalcification. The stage of microcalcification renders the plaque unstable and more predisposed to rupture, making its identification an interesting surrogate marker for detecting "vulnerable" plaques. NaF-PET was shown to be able to identify microcalcification in atherosclerotic plaques in a consistently different pattern of uptake to the macroscopic calcium observed on computer tomography (CT) [22] (Figure 1).

Different groups have proposed that NaF uptake could be used as a maker of cardiovascular risk by demonstrating its correlation with various validated clinical scores for cardiovascular disease burden [23–26]. Our group showed that in a high cardiovascular risk population, NaF atherosclerotic plaque uptake was related to the burden of cardiovascular risk factors and thoracic fat volume, but there was no association between coronary uptake and calcium score [27].

In the context of acute coronary syndrome patients, the culprit plaques associated with infracted myocardium were shown to display greater NaF uptake than "non-culprit" plaques [28].

In perhaps the most significant clinical study with NaF-PET, patients with known coronary artery disease underwent NaF-PET/CT and were followed up for fatal or nonfatal myocardial infarction over 42 months. Total coronary NaF uptake predicted MI independently of age, sex, risk factors, segment involvement and coronary calcium scores, presence of coronary stents, coronary stenosis, REACH and SMART scores, the Duke coronary artery disease index, and recent myocardial infarction [29].

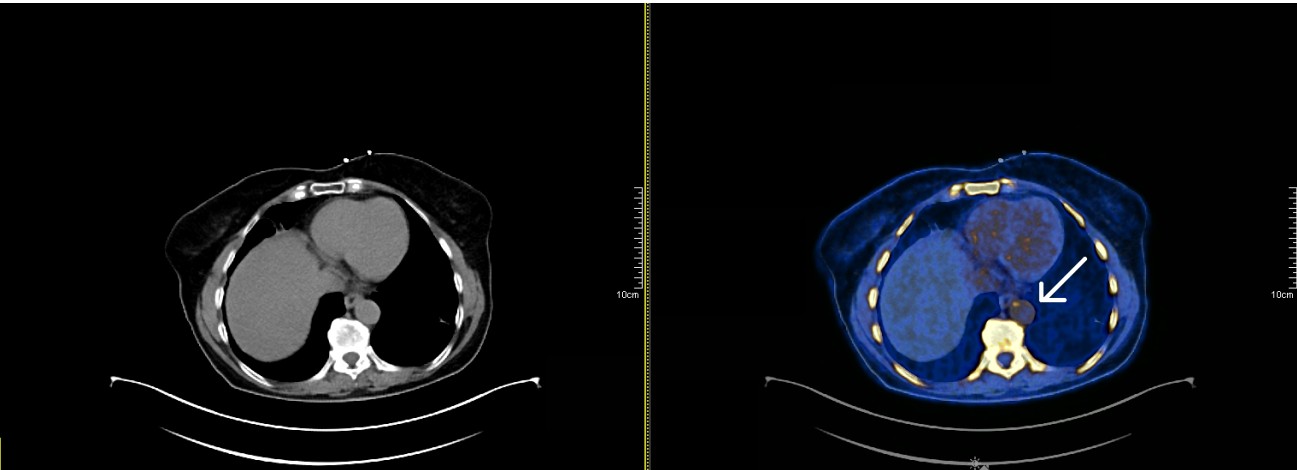

**Figure 1.** Fusion 18F-NaF-PET-CT images, depicting NaF uptake in the descending aorta (arrow, right picture). In the left picture, corresponding to the raw CT image, one can observe that the NaF uptake matches an area without macroscopic calcification. NaF was injected 60 min before image acquisition. Image source: Institute for Nuclear Sciences Applied to Health (ICNAS).

## 3. Perfusion Imaging

In perfusion imaging, stress and rest myocardial perfusion image sets are compared in order to determine the presence, extent, severity, and reversibility of stress-induced perfusion defects [30].

Similarly to single-photon emission computerized tomography (SPECT), PET can provide a visually graded qualitative assessment of relative perfusion defects (Figure 2). However, PET also allows quantitative assessment of myocardial blood flow (MBF) during the different stages of the exam, which allows the calculation of myocardial flow reserve (MFR)—the ratio of MBF in a hyperemic state and at rest (Figure 3). Quantification of absolute MBF and MFR appears to add prognostic value to the qualitative assessment [31–33].

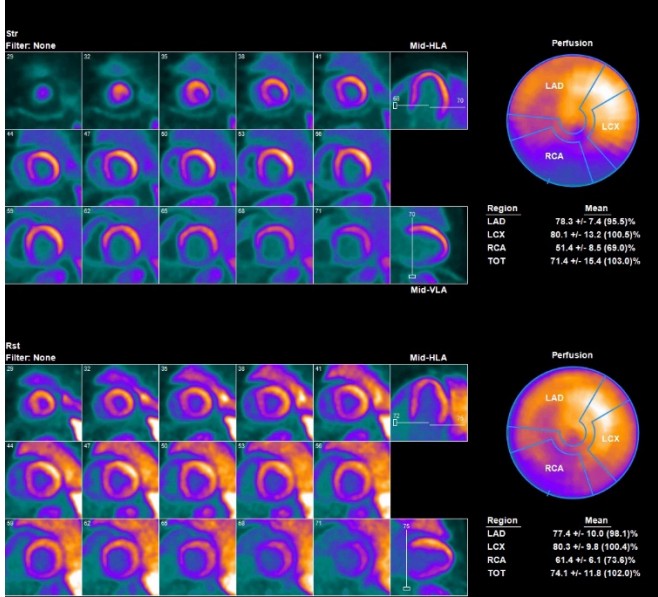

**Figure 2.** Assessment of myocardial perfusion with 13N-ammonia-PET. Top rows represent stress acquisition and lower rows rest acquisition. Myocardial perfusion is markedly decreased in the inferior wall during stress, compatible with ischemia in this territory. Image source: Institute for Nuclear Sciences Applied to Health (ICNAS).

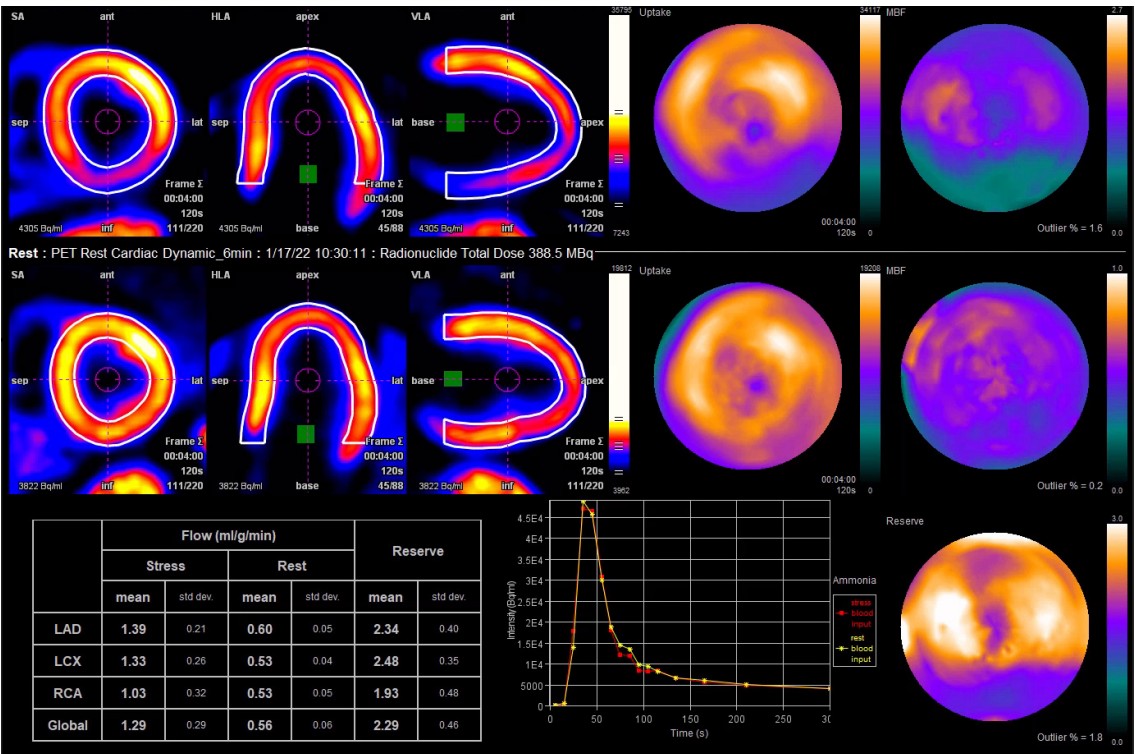

**Figure 3.** Quantitative assessment of myocardial perfusion with 13N-ammonia-PET of the same patient represented in Figure 2. Note that myocardial flow during stress is decreased in the RCA territory. LAD—left anterior descending artery, LCX—left circumflex artery; RCA—right coronary artery. Image source: Institute for Nuclear Sciences Applied to Health (ICNAS).

### 3.1. Imaging Myocardial Perfusion with Rubidium-82 (Rb)

Rb myocardial uptake is proportional to MBF, as was demonstrated for the first time more than half a century ago [34]. It is the most widespread tracer used for PET myocardial perfusion imaging (MPI), since it does not require a cyclotron on-site. It has a half-life of 78 s and an extraction fraction in comparison to MBF of around 60% [35].

Compared with conventional MPI with SPECT, Rb-PET showed improved image quality, higher diagnostic accuracy, less radiation dose to patient and staff as well as rapid examinations time [36]. It has also shown better sensitivity for the detection of multivessel disease, which in cases of balanced ischemia may present as a false negative in SPECT [35]. A recent meta-analysis compared the diagnostic performance of cardiac magnetic resonance (CMR), SPECT, and PET imaging for the identification of CAD and concluded that both CMR and PET were superior to SPECT [37]. However, a randomized study comparing the clinical effectiveness of pharmacologic SPECT and PET MPI in symptomatic CAD patients ($n$ = 322) showed no significant differences between the two groups in subsequent rates of coronary angiography, coronary revascularization, or health status at 3-, 6-, and 12-month follow-ups [38].

In a cohort of 16,029 consecutive patients undergoing Rb rest–stress PET MPI, patients with higher degrees of ischemia had a survival benefit from early revascularization [39].

### 3.2. Imaging Myocardial Perfusion with 13N-Ammonia

13N-ammonia is uptaken by the cardiomyocytes, after which it is irreversibly trapped inside the cell. It has a half-life of 9.8 min and an extraction fraction in comparison to MBF of around 80% [35].

There are no studies that directly compare the diagnostic accuracy of 13N-ammonia PET with SPECT, although it may have higher sensitivity relative to Rb due to its higher

myocardial extraction [35]. In unselected patients with indication for MPI, cardiac perfusion findings in 13N-ammonia PET were strong predictors of long-term outcome [40].

### 3.3. Imaging Myocardial Perfusion with 15O-Water

Oxygen-15-labelled water is a freely diffusible and metabolically inert tracer, and is considered the best tracer for quantitative studies [41]. It has a half-life of 2.4 min and an extraction fraction in comparison to MBF of around 95% [35].

A prospective clinical study involving 208 patients with suspected CAD who underwent CCTA, technetium 99 m/tetrofosmin–labeled SPECT, and 15O-water PET with examination of all coronary arteries by fractional flow reserve, revealed that PET exhibits the highest accuracy for diagnosis of myocardial ischemia [42]. Although not FDA approved and mainly used in research, routine clinical use of 15O-water PET with a bedside generator and infusion solution has proven to be reliable and efficient [43].

## 4. Viability Imaging

As previously described, one important objective of perfusion imaging is differentiating between ischemic myocardial tissue and myocardial scar. This task may be challenging with conventional perfusion imaging in the presence of myocardial hibernation. Hibernated myocardial tissue is in a state of metabolic downregulation in response to chronic or repetitive ischemia that can potentially be recovered with coronary revascularization [30].

### Imaging Myocardial Viability with 18F-Fluorodeoxyglucose (FDG)

The biochemical properties of this tracer have been previously described. Demonstration of preserved glucose metabolism by FDG is a marker of myocardial viability. While reduced perfusion combined with reduced glucose metabolism suggests scarred myocardium, reduced perfusion combined with preserved or increased metabolism (mismatch) suggests hibernating myocardium.

From a theoretical point of view, management of hibernated myocardial tissue should be straightforward—revascularization. However, some studies have questioned this assumption.

In the PARR-2 (PET and recovery following revascularization) trial, patients with severe left ventricular dysfunction were randomized to revascularization decision managed by FDG-PET versus standard care. Management by FDG-PET did not result in reduction of death, MI, or recurrent hospital stay at 1 year compared with standard management [44]. However, in a post hoc analysis of a group of patients belonging to a more experienced center with ready access to FDG-PET and integration with imaging, heart failure, and revascularization teams, a significant reduction in cardiac events was observed in patients with FDG-PET-assisted management [45].

In the viability sub study of the STICH (surgical treatment for ischemic heart failure) trial, there was no significant association between myocardial viability and outcome on multivariable analysis [46].

In light of the described evidence, the most recent European Guidelines on myocardial revascularization give the use of noninvasive stress imaging for the assessment of myocardial ischemia and viability in patients with heart failure and coronary heart disease before the decision on revascularization a Class IIB recommendation [47].

## 5. Final Remarks

PET is a powerful tool for the diagnosis of coronary heart disease. Compared with the most commonly used nuclear exam, SPECT, it has a higher diagnostic accuracy and the addition of quantitative information yields incremental prognostic value. Cardiac PET can comprehensively assess all aspects of coronary heart disease, from coronary atherosclerotic plaque to the myocardial tissue characterization. However, how this information can be transferred to real-world practice and help to guide decision making is still a hot topic of research.

**Author Contributions:** Conceptualization, J.d.A.; data collection, J.d.A. and S.M.; data processing, J.d.A. and S.M.; writing—original draft preparation, J.d.A. and S.M.; writing—review and editing, L.G. and M.F. All authors have read and agreed to the published version of the manuscript.

**Funding:** This research received no external funding.

**Institutional Review Board Statement:** Not applicable.

**Informed Consent Statement:** Not applicable.

**Data Availability Statement:** Not applicable.

**Conflicts of Interest:** The authors declare no conflict of interest.

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
