# Peer review of "Positron Emission Tomography in Coronary Heart Disease"

_applsci, doi:10.3390/app12094704_

Round 1

Reviewer 1 Report

This short review is very interesting.

I have the 3 following comments:

1- May I suggest to show the assessment of myocardial perfusion with 13N-ammonia-PET and Rubidium-82 (Rb) in one patient and/or one animal in order to compare these two imaging modalities.

2- The presentation of the References section is heterogeneous. Please correct.

5]Tarkin JM, Ćorović A, Wall C, Gopalan D, Rudd JH. Positron emission tomography imaging in cardiovas-cular disease. Heart. 2020;106(22):1712-1718. 

[6] Andrews JPM, Fayad ZA, Dweck MR. New methods to image unstable atherosclerotic plaques. Athero-sclerosis. 2018;272:118-128. 

[7] Hyafil F, Cornily JC, Rudd JHF, Machac J, Feldman LJ, Fayad ZA. Quantification of inflammation within rabbit atherosclerotic plaques using the macrophage-specific CT contrast agent N1177: a comparison with 18F-FDG PET/CT and histology. J Nucl Med. 2009;50(6):959-965. 

[8] Duivenvoorden R, Mani V, Woodward M, et al. Relationship of serum inflammatory biomarkers with plaque inflammation assessed by FDG PET/CT: the dal-PLAQUE study. JACC Cardiovasc Imaging. 2013;6(10):108794. 

[9] Yoo HJ, Kim S, Hwang SY, et al. Vascular inflammation in metabolically abnormal but normal-weight and metabolically healthy obese individuals analyzed with (1)(8)F-fluorodeoxyglucose posítron emission to-mography. Am J Cardiol. 2015;115(4):5238. 

3- Please insert a space after the term "with" in the following sentence "Imaging myocardial perfusion withRubidium-82 (Rb)" 

Reviewer 2 Report

this review addresses the issue of PET applications in cardiovascular imaging. It represents a general overview concerning technical aspects and potential clinical applications.

This approach lacks in details because doesn’t produces enough useful information neither on instrumentation neither on clinical issue.

In particular the distinction of all paragraphs in two sections (general and clinical impact) doesn't make any sense. The sections have to be unified.

Recent articles and one review have been analysed the relationship between immune-regulation and glucose metabolism. These papers have to be considered for the revision of FDG section.

FIG1. the image should recomposed to emphasise the relation with calcium content of the plaque detected by CT. Please report the time from NAF injection

THE PARAGRAPH SHOULD PROVIDE A DEFINITIVE CLINICAL INDICATION WHEN POSSIBLE.

PLEASE INSERT A LARGER DISCUSSION CONCERNING THE Rb ADVANTAGES OVER TRADITIONAL MPI.

Round 2

Reviewer 2 Report

the quality of paper has been improved